# US Swine Industry Stakeholder Perceptions of Precision Livestock Farming Technology: A Q-Methodology Study

**DOI:** 10.3390/ani13182930

**Published:** 2023-09-15

**Authors:** Babatope E. Akinyemi, Faical Akaichi, Janice M. Siegford, Simon P. Turner

**Affiliations:** 1Department of Animal Science, Michigan State University, East Lansing, MI 48824, USA; siegford@msu.edu; 2Rural Economy, Environment and Society Department, Scotland’s Rural College (SRUC), Edinburgh EH9 3JG, UK; faical.akaichi@sruc.ac.uk; 3Animal and Veterinary Sciences Department, Scotland’s Rural College, Easter Bush, Edinburgh EH25 9RG, UK; simon.turner@sruc.ac.uk

**Keywords:** precision livestock farming, Q-methodology, stakeholder, pork industry

## Abstract

**Simple Summary:**

Understanding the views and concerns of precision livestock farming (PLF) held by the key players in the US swine industry is particularly important for its widespread acceptability. Using a Q-methodology approach following initial individual in-depth interviews 6 months earlier, we found that while most stakeholders view PLF as good and beneficial to pigs and humans, some expressed concerns about its limitations and pitfalls. Specifically, stakeholders who are optimistic about PLF think it should augment but not replace effective management. They also think it will improve pig health, welfare, and make pig keepers’ jobs simpler, safer, and better. However, stakeholders who are concerned about PLF view it as a poor proxy for farmers to keep an eye on the herd, see issues, and resolve them. They also believe PLF does little to lessen the negative effects of swine farming on the environment. We contend that competing perceived benefits and concerns about PLF may hinder widespread adoption.

**Abstract:**

This study used the Q-methodology approach to analyze perceptions of precision livestock farming (PLF) technology held by stakeholders directly or indirectly involved in the US swine industry. To see if stakeholders’ perceptions of PLF changed over time as PLF is a rapidly evolving field, we deliberately followed up with stakeholders we had interviewed 6 months earlier. We identified three distinct points of view: PLF improves farm management, animal welfare, and laborer work conditions; PLF does not solve swine industry problems; PLF has limitations and could lead to data ownership conflict. Stakeholders with in-depth knowledge of PLF technology demonstrated elevated levels of optimism about it, whereas those with a basic understanding were skeptical of PLF claims. Despite holding different PLF views, all stakeholders agreed on the significance of training to enhance PLF usefulness and its eventual adoption. In conclusion, we believe this study’s results hold promise for helping US swine industry stakeholders make better-informed decisions about PLF technology implementation.

## 1. Introduction

Global meat consumption has increased significantly over the past few decades, with pork and poultry showing the most substantial growth in consumption rates [1]. In fact, the number of animals kept on the farm has grown substantially, while the number of farmers has dwindled [2]. As a result, each farmer has considerably larger herds, which makes it challenging for them to consistently monitor every animal in such large groupings in a way that assures each animal’s welfare while minimizing environmental impact. Specifically in the swine industry, the low-profit, high-risk [3] nature of pork production further exacerbates this situation.

Precision Livestock Farming (PLF) is a management system that involves using technology to continuously collect and process data from individual animals [2]. This is significantly different from other approaches that monitor animal welfare by human experts scoring animal-based indicators on an infrequent basis. The application of PLF technology has recently been proposed as one of the ways to help address public concerns transparently and objectively [4]. According to the proponents of PLF, these technologies can assist with many issues, including farmer profitability, environmental concerns, and animal welfare, and might even be able to improve all these issues at once [5].

Although PLF holds many promises, it also raises many concerns. As PLF evolves, further farm consolidation is possible since only those with the financial wherewithal to invest in PLF may benefit from it [6,7]. The loss and “de-skilling” of agricultural employment is another issue associated with the use of PLF in animal agriculture [8]. As livestock monitoring and care become more automated, PLF will need less labor and those jobs will not require many of the skills that are currently needed [7]. 

According to Werkheiser [5], there are other broader societal implications of PLF. Firstly, there is a worry that resources are being allocated to develop technology that could be used for monitoring humans as well [5]. Secondly, there is a risk that this technology becomes normalized, which could decrease resistance to its use, both in terms of legal regulations and public perception [5].

Hitherto, very few studies have investigated stakeholders’ perceptions of PLF—exceptions are Giersberg and Meijboom [9], Klerkx and colleagues [10], and Pfeiffer and colleagues [11]. Specifically, there is no empirical study on the suitability of PLF technologies for different stakeholders within the US swine industry, and the perspectives of various stakeholders on the implementation of these technologies and the utilization of PLF data are not yet known [10]. Moreover, there is a paucity of information on the benefits PLF offers, as well as the concerns and limitations associated with its implementation [9]. Consequently, there is a significant knowledge gap that must be addressed to fully comprehend the broader implications and viability of PLF technologies within the US swine industry.

This study examined US swine industry stakeholders’ perceptions of and needs for PLF across the pig production system ranging from areas of genetic improvement to welfare certification. We adopted Q-methodology to quantify swine stakeholders’ perceptions of PLF technology. The growing recognition that the opinions of stakeholders involved in the development and usage of technology must be considered for decisions to be recognized as legitimate is a major factor in Q-methodology’s rising popularity in technology-related studies [12,13,14].

## 2. Materials and Methods

The Q-methodology approach combines quantitative and qualitative techniques to examine people’s subjective beliefs scientifically and enables researchers to statistically identify and categorize opinions on a given topic [15]. In a Q-study, an individual is presented with a set of statements about a given topic and then asked to rank-order the statements. In our study, participants ranked statements from “Most Like What I Think” to “Least Like What I Think “ using the Q-grid in Figure 1. This operation is referred to as Q-sorting.

The Q-methodology procedure shown in Figure 2 is briefly described here. The first step in the Q-method entailed defining and building a concourse. The concourse refers to the collection of subjective statements (from the Latin concursus, meaning “a running together” as when ideas run together in thought) [16]. Concourses are statements describing the subjective views, opinions, and arguments put forward by individuals, stakeholders, professionals, and scientists about a particular topic (in this study, PLF). Meanings attributed to the concourse are inherently social and contextual; hence, it is assumed that stakeholders would inevitably draw from the ongoing PLF social discourse in constructing and articulating their experiences and perceptions of PLF [16].

The second step was the development of the Q-set, which was a subset of statements drawn from the larger concourse, and it was this set of statements that was presented to participants in the form of a Q-sort. The Q-set for this study was developed from in-depth interviews reported by Akinyemi and colleagues [17]. The Q-set comprised subjective statements on PLF made by 12 individuals. These individuals were purposefully chosen to reflect the diversity of PLF perspectives across the swine industry ranging from pig producers to pork consumers. The stakeholders were domain experts in swine welfare, swine veterinary medicine, animal welfare auditing, animal health regulation, technology development, agricultural engineering, animal care, and compliance, animal breeding and genetics, and consumer advocacy. The demographic characteristics of the participants are shown in Table 1. 

The 30 Q-set statements that were chosen for the Q-sorting exercise were derived from 55 subjective statements on PLF that were obtained from the 12 interviews. Ten main themes emerged from the 55 statements after careful analysis and categorization. These themes were not a strict categorization; rather, they represent coverage of the most relevant topics related to stakeholders’ views on PLF. After several rounds of discussion among authors, redundant statements were either rephrased clearly, closely similar themes were merged, and vague statements were deleted. At the end of the process, the initial 55 statements were refined into a more comprehensive Q-sample composed of 30 statements. We also felt these 30 statements would reduce the cognitive burden of ranking 55 statements in 30 min on the participants. We ensured that the Q-samples were composed of statements that were “natural” in the language of stakeholders and “comprehensive” in their PLF views [18] to provide them with the opportunity to best express their personal views of PLF [19].

Gender and level of education were omitted to preserve participants’ anonymity. The third step in this process entailed choosing the Q-participants which are referred to as the ‘P-set’. In our case, the selection of the P-set was not performed at random; rather, these participants were 11 of the 12 original stakeholders who informed the Q-set and had been chosen to reflect diverse PLF perspectives across the US swine industry. This sample size is considered appropriate for this type of study [20,21,22,23] since P-set is expected to be composed of individuals who are theoretically relevant to the research question [24,25]. We deliberately made the decision to use the same group of stakeholders to develop the concourse underlying the Q-set and to participate in ranking the Q-set statements for two reasons. First, we wished to see how US swine industry stakeholders would view their own statements as well as statements from others in the group to examine how the stakeholders as a whole were similar to or different from one another. Furthermore, we wanted to examine whether their individual perspectives about PLF, which is a rapidly changing landscape, had changed in the 6 months between their original interviews and the Q-sort exercise.

The actual Q-sorting was conducted in step four. During Q-sorting, each participant ranked the 30 Q-statements into a Q-grid (see Figure 1). The Q-grid created a forced quasi-normal distribution using a pre-determined pattern grid with a scale labeled “More Like What I Think”, “Neutral”, and “Least Like What I Think”. The Q-sort was concurrently administered to a group of 9 persons in a conference room while 2 joined online via Zoom. The instructions (see Appendix B) for the Q-sort exercise were read and explained to all the participants who were also given the chance to ask clarifying questions. The Q-sorting lasted for approximately 30 min. During the Q-sorting, participants were given a set of 30 index cards (9 persons in the room were given printed index cards while the 2 participants on Zoom were given electronic index cards). Each index card had one of the 30 Q-set statements written on it and randomly numbered. They were given the Q-sort grid with the instructions, and they were first asked to sort the cards into three piles of “More Like What I Think”, “Least Like What I Think”, and “Neutral (statements for which participants had no opinion)”, based on their opinion of PLF. Next, they ranked the 30 statements on the 30-item forced normal distribution ranging from +5 (More Like What I Think) to −5 (Least Like What I Think). Participants began by taking the statements they grouped as “More Like What I Think” and chose the one statement that was “most like what I think” to place on the worksheet in the “+5” column on the far right. They were then asked to place the next two statements “More Like What I Think” into the “+4” column (in no particular order) until all statements from their “More Like What I Think” pile were placed on the Q-sort grid. The process was repeated with “Least Like What I Think” statements, starting with a statement in the “−5” column. When the participants were satisfied with how they had arranged the concourse statements on the Q-sort grid, they wrote the number on the statement card inside the corresponding squares in the Q-sort grid. A completed Q-sort grid is shown in Section E.1, Section E.2 and Section E.3. Finally, participants provided basic demographic information (Table 1). The entire process lasted for about 30 min.

In step five, the 11 completed Q-sorts were analyzed using a by-person correlation and factor analysis approach using PQMethod free online software [26]. Factor analysis is the practice of condensing many variables into just a few so that research data are easier to work with. The idea is to identify and work with deeper factors driving the underlying concepts within data and then uncover and work with these rather than the lower-level variables that flow from them. Factor analysis is also sometimes called “dimension reduction analysis”. With factor analysis, data “dimensions” can be reduced into one or more “super-variables”, also known as unobserved variables or latent variables. For more information on factor analysis, please see Mulaik [27]. 

In this study, factor analysis was used to identify correlations (see Table 2) between the 11 Q-sorts and to reduce the many ‘viewpoints’ of PLF expressed through the sorting pattern down to 3 factors. These 3 factors represent unique ways of thinking about PLF among the 11 participants who shared the same factor. These factors are then interpreted as perspectives. In Q-methodology, ‘viewpoints’ are defined as a common set of perceptions profiles from participants that form a cluster of correlations [24]. By default, the analysis produced eight unrotated factors, which accounted for 95% of the total variance in the Q-sorts. Second, a varimax rotation was used to identify a small number of factors with significant factor loadings.

Before factors were extracted, we had to decide the number of factors to be extracted and retained as being significant via the PQMethod software. Following Brown’s [25] guidelines on factor stability, distinctness, clarity, and simplicity, we determined the number of factors to rotate based on (1) the eigenvalue being equal to or greater than one and (2) each factor should have at least two significant factor loadings in the unrotated factor matrix. Two of the eight factors met both criteria (see Table 2). The third factor was included in the rotation because its eigenvalue was close to one (0.94) and satisfied Humphrey’s Rule, which states that a factor is significant if the absolute value of “the cross-product of the two highest loadings exceeds the standard error” [19]. After rotating the three factors (see Table 2), we found that at least two participants were loaded on each factor, the eigenvalue of the three factors was higher than one, and the three factors accounted for 73% of the total variance.

In the last step, we focused the interpretations of the factor analysis on the rotated solution factor Q-sort values. The unrotated factor solution is presented in Appendix F. For each factor, we considered the statements with the highest and the lowest Q-sort values (see Table 3). The highest-ranking statements (i.e., Q-sort values that are between +3 and +5) indicate what participants who loaded on the factor of interest think of PLF and least like what participants think of PLF for the lowest-ranking statements. We also paid close attention to the distinguishing statements (see Appendix D) which are the statements that were found to be significantly different among factors. The consensus statements (Appendix D) were also found to be helpful in identifying the PLF-related issues that all participants could agree on even though they loaded on varied factors.

## 3. Results and Discussions

The by-person correlation matrix in Table 2 shows the correlation between the 11 participants who participated in this study. A perfect positive and negative correlation (r) corresponds to +1, and −1, respectively. As a rule of thumb, correlations are generally considered to be statistically significant if they are 2 to 2.5 times the standard error (i.e., between 0.44 and 0.56 irrespective of sign but the correlation coefficients considered significant in this study are r = 56 and above) [16]. Hence, participant P6 in the first row/column was statistically significantly and positively correlated with P8 (r = 75), P7 (r = 73), P10 (r = 64), and P5 (r = 60), but weakly and negatively correlated with P3 (r = −38) and P4 (r = −29). In addition, P6 moderately positively correlated with P1 (r = 58) and P2 (r = 50); however, its positive correlation with P11 (r = 24) was not substantial. Moreover, participants P5 and P7, P8 and P10, and P2 and P10 were strongly positively correlated with each other with the correlation coefficients of r = 68, r = 64, and r = 62, respectively. It is worth noting that the statistics associated with Q are not intended as a substitute for the apparent fact that the correlation is suffused with subjectivity, each Q-sort being a transformation of a person’s opinion, and the coefficients merely indicate the degree of similarity or dissimilarity in perspective.

Presented in Table 4 are the factor loadings from the PCA. In essence, PCA uses the correlation matrix shown in Table 2 to determine how many fundamentally distinct Q-sorts are present. Q-sorts that are highly correlated with one another may be considered to have a family resemblance, with members within one family being strongly correlated with one another but not with those of other families. The number of various families (factors) is revealed via PCA. Therefore, the number of factors is entirely reliant on the actual performance of the Q-sorters and is simply empirical. In this study, the factors represent different perspectives of PLF, with those individuals sharing a common perspective defining the same factor. As previously noted, each factor represents different perspectives or conceptualizations of PLF. Factor 1 perspective of PLF is shared by seven participants, namely P1, P2, P5, P6, P7, P8, and P10. Factor 2, however, is a manifestation of a strong bipolarity between P3 and P4 on the positive side of the pole and P1, P5, and P7 on the negative side of the pole. Factor 3 represents a single perspective of PLF shared by participants P9 and P11. A sort is significantly associated with a factor at *p* < 0.01 statistical significance if the absolute value of the factor loading is greater than 2.58 ÷ √ (the number of statements used in the Q-sort)), which is equal to 0.47 in this study [25]. Based on the Q-sort association with each factor, P1, P2, P5, P6, P7, P8, and P10 are correlated with Factor 1. However, P3 and P4 are correlated with Factor 2 whereas P9 and P11 are correlated with Factor 3.

Based on the results in Table 3 and Appendix C, Appendix D and Appendix E, three perspectives were identified. Factor 1 represents Perspective 1: “PLF improves management, animal welfare, and laborer work conditions”; Factor 2 represents Perspective 2: “PLF does not solve the problems”; and Factor 3 represents Perspective 3: “PLF has limitations and could lead to data ownership conflict”. Table 3 shows the bipolar ranked factor score of five statements of “more like what I think” of PLF (corresponding to the positive factor scores) and “least like what I think” of PLF (corresponding to the negative factor scores). The higher positive factor scores imply strong agreement with the statement whereas higher negative factor scores mean strong rejection of the statement. For example, the statement “PLF should be a supplement to good management and not replace good management” with the highest factor score of +5 indicates what participants sharing Perspective 1 strongly think of PLF.

The two statements “PLF will improve pig health” and “PLF makes the pig caretaker’s job easier, safer, and better” with factor score +4 also reflect what Perspective 1 participants think of PLF. This group strongly rejects the following statements: “PLF’s data often cannot be used within existing management practices” (−5), “PLF can disconnect caretakers from pigs and make them less caring about pigs” (−4), and “PLF can be used without any form of training” (−4).

Conversely, Perspective 2 participants strongly think “PLF can disconnect caretakers from pigs and make them less caring about pigs” (+5), “PLF is a poor proxy of a farmer looking over the herd, detecting problems and addressing them” (+4), and “PLF technology does not minimize the environmental impact of swine farming” (+4). They rejected the notion that “PLF will improve pig welfare “(−5), “PLF will improve pig health (−4), and “PLF can be used without any form of training” (−4).

Participants sharing Perspective 3 strongly think “PLF usage is limited by poor internet connection on most swine farms” (+5). They share the concern that” PLF usage can lead to data privacy and data ownership conflicts” (+4), and that “PLF often generates data that cannot be used within existing management practices” (+4). They strongly rejected the idea that “PLF technology can be used without any form of training” (−5). They also disagreed with the notion that “PLF technology will address public concerns and increase consumer trust in pork production” (−4), and that “PLF is a poor proxy of a farmer looking over the herd, detecting problems, and addressing them” (−4).

The 30 Q-set statements and the sources of each statement are shown in Appendix G. As expected, all stakeholders tend to agree with their own statements as well as statements from other stakeholders sharing similar roles or relationships with PLF. The PLF technology developers and large-scale farmers generally agreed with each other but disagreed sharply with statements from potential off-farm PLF users. Their Q-sorting was consistent with the views of PLF they expressed earlier during the interview conducted six months before the Q-sort exercise.

For example, the statements “PLF will improve pig health” and “PLF will improve pig welfare” were made by technology developers and large-scale farmers during their individual interviews six months earlier and were later ranked highly with factor scores of (+4) and (+3) (Table 4), respectively, indicating these statements were more like what this group thinks of PLF. However, they rejected the statements “PLF can be used without any form of training” (−4) and “PLF can disconnect caretakers from pigs and make them less caring about pigs” (−4), which were statements made by potential off-farm users during individual interviews six months earlier.

Likewise, the potential off-farm PLF user group held on to the views of PLF previously shared during the individual interview six months earlier and ranked the statements “PLF technology can disconnect caretakers from pigs and make them less caring about pigs” (+5) and “PLF technology is a poor proxy of a farmer looking over the herd, detecting problems, and addressing them” (+4) highly. They rejected the two statements from the PLF technology developer and farmer’s group that “PLF will improve pig health” (−4) and “PLF will improve pig welfare” (−5) and ranked them very low.

Perspective 1: PLF improves management, animal welfare, and labor work conditions.

Seven stakeholders (P1, P2, P5, P6, P7, P8, and P10) shared Perspective 1. They included three technology developers, two swine veterinarians, a swine farmer, and an animal care and compliance specialist. This group is distinctly different from those with Perspectives 2 and 3 in that participants who shared this view were optimistic and enthusiastic about PLF and were less worried about its limitations and concerns. This group emphasized PLF benefits such as improvements to swine farm management, pig health, and animal welfare.

For this group, it was important that PLF should supplement good animal husbandry and not replace caregivers. According to [28,29], existing swine production management practices require skilled workers to work long hours in a challenging environment in ways that can both harm the workers’ mental and physical health as well as increase the animals’ biosecurity risks. Therefore, PLF can improve laborers’ work conditions through automated non-invasive monitoring and management [30,31] that allows targeted, problem-solving work rather than repetitive manual labor. In addition, in view of the anticipated labor shortage in the swine industry in the near future [32], PLF technology that improves laborer work conditions might serve as an incentive for retaining and attracting farm workers.

Perspective 1 shares the findings of Benjamin and Yik [33] that PLF will help in controlling diseases by closely monitoring animals which allows farmers, veterinarians, and others to ensure pigs’ welfare. Further potential to track animals may assist with the traceability of pork products through the production and distribution value chain. This is particularly important in view of the low-profit margins in swine production and the increasing pressures related to the responsible sourcing of pork. These pressures may create a compelling need and impetus for swine farmers to adopt cost-efficient technologies that can monitor the actual state of pigs to provide transparency, traceability, and evidence of improved welfare needed to meet consumer demands.

Most of the Perspective 1 stakeholders had practical knowledge of PLF which may have informed their beliefs about its value and potential and explained their high optimism about PLF’s benefits to the swine industry. It is noteworthy that this group shared the views of Perspective 2 because they do not see PLF as disconnecting caretakers from pigs and making them less caring, or that PLF data are not usable within existing farm management practices, and as displacing labor and reducing job opportunities in the swine industry.

Perspective 2: PLF does not solve the problems.

Two stakeholders (P3 and P4) share Perspective 2. These stakeholders had no prior direct knowledge of PLF. This perspective is dissimilar to the first perspective in that it is less optimistic about PLF technology. These respondents thought PLF would not fix the problems facing the swine industry. This group was worried about the ways PLF could change the relationships between farmers and their livestock by eliminating the bond between farmers and their animals that develops as they spend time directly assessing their well-being [34]. They were particularly concerned that animal caretakers may become under- or over-reliant on PLF technology and spend less (quality) time with their pigs which could cause a loss of animal-oriented husbandry skills.

Moreover, Perspective 2 disagreed with Tullo and colleagues [35] and Lovarelli and colleagues’ [36] claims that PLF can promote sustainability by enabling farmers to feed their animals with more precision, thereby avoiding waste [37], thus increasing both environmental and economic sustainability for the farm. On the contrary, they thought PLF technology does not reduce the negative externalities of swine farming but digitizes the swine industry, giving it a more artificial and unnatural outlook. Stakeholders in the Perspective 2 group felt that PLF technology should be used with effective management, not as a replacement. Finally, respondents favoring Perspective 2 were pessimistic about the claims that PLF could improve pig welfare and pig health. There was also skepticism among this group that PLF will allay public concerns, boost consumer confidence in pork production, and enable consumers to independently verify welfare certification claims. However, they agreed with Perspectives 1 and 3 that PLF technology cannot be used without training.

Perspective 3: PLF limitation and data ownership conflict.

Two stakeholders (P9 and P11) shared Perspective 3. This group was concerned about limitations, data ownership, and conflict issues associated with PLF usage. Echoing previous studies [38,39,40,41], this group considered poor internet connection a major problem on most swine farms in Australia, Europe, and the USA. Although internet connectivity is becoming widespread on farms across the world, the connection is sometimes still unreliable and too slow to carry large amounts of data. Hence, implementing PLF technologies on commercial farms poses challenges because transferring data to the cloud might not be optimal or even feasible in all circumstances. Edge computing, which enables data to be processed at the farm level rather than being sent over the internet, is one potential solution to these challenges [42].

Another issue associated with PLF emphasized by the Perspective 1 and 3 groups was data privacy and data ownership conflicts. A previous study by Neethirajan and Kemp [43] emphasized the need to ensure data privacy and security in precision livestock farming. According to Wolfert and colleagues [44], data collection on or from farms is currently limited because farmers prioritize privacy. To address this, new advances in machine learning are being developed that utilize privacy-preserving data exchange systems. Successful commercialization of PLF technologies has been hampered by a lack of open access to and proprietary control in data ownership held by a few commercial companies [30].

In contrast to the Perspective 1 group, the Perspective 3 group believed PLF data cannot be utilized within current management practices and that it would take substantial modification to adapt current production system processes to fully utilize PLF data. Stakeholders with Perspective 3 concurred with Morrone and colleagues [4] that PLF technologies are too expensive and may reduce the profitability of farming operations. In line with Perspectives 1 and 2, Perspective 3 stakeholders saw the need for training to effectively utilize PLF technology, echoing [38]. They rejected the claim that PLF technology would allay public worries and boost consumer confidence in pork production. They neither saw PLF technology as a solution to labor problems nor as displacing labor and reducing job availabilities in the swine industry. Likewise, they did not see PLF technology as a poor proxy for a farmer detecting problems and addressing them.

## 4. Conclusions

This study investigated how persons with varied involvement in the swine industry saw precision livestock farming technology. Three distinct viewpoints were identified by this study using Q-methodology: PLF improves farm management, animal welfare, and labor work conditions; PLF does not solve swine industry problems; and PLF has limitations and could lead to data ownership conflicts. The results uncovered the diversity of perspectives held by swine stakeholders about PLF technology and provided common descriptions of viewpoints.

We noticed dissimilarities in the opinions of stakeholders with direct and indirect PLF knowledge. Stakeholders with direct and advanced knowledge of PLF (i.e., technology developers and large-scale farmers) consistently displayed high optimism about PLF technology over a six-month period whereas stakeholders with indirect and limited knowledge (i.e., potential off-farm PLF users) tended to be skeptical of PLF claims over the same period. Nevertheless, these perspectives were also influenced by their employment role, as those most familiar with PLF were more actively involved in its development. This study emphasized the significance of training to enhance perceptions of PLF’s usefulness and its eventual implementation.

## 5. Limitations of This Study

Although the 11 individuals that constituted the P-set for this study meet Brouwer’s [45] recommendation of 10 to 40 people and these individuals perform different roles across the swine industry, there was some imbalance in the perspectives represented. Veterinarians, technology developers, and large-scale farmers dominated the study population, while government officials and consumers were least represented, whereas smallholder farmers, nutritionists, NGOs, and extension agents were not represented at all. These limitations should be taken into consideration when interpreting findings from this study. Future studies confirming the three dominant perspectives in underrepresented populations are needed. In addition, given that the current study focused on the US swine industry, generalizing findings from this study to other regions of the world should be carried out with caution. Nevertheless, future research may benefit from implementing similar studies in other regions of the world.

## Figures and Tables

**Figure 1 animals-13-02930-f001:**
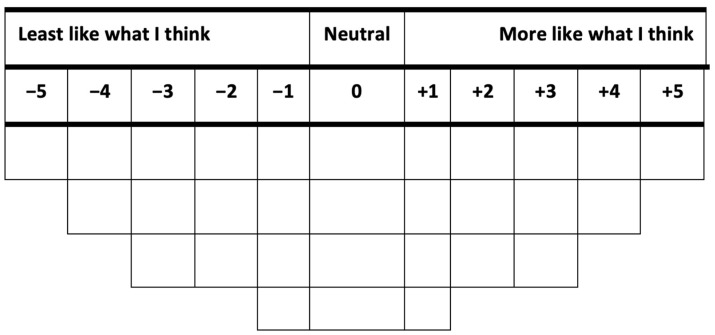
Q-sort grid.

**Figure 2 animals-13-02930-f002:**
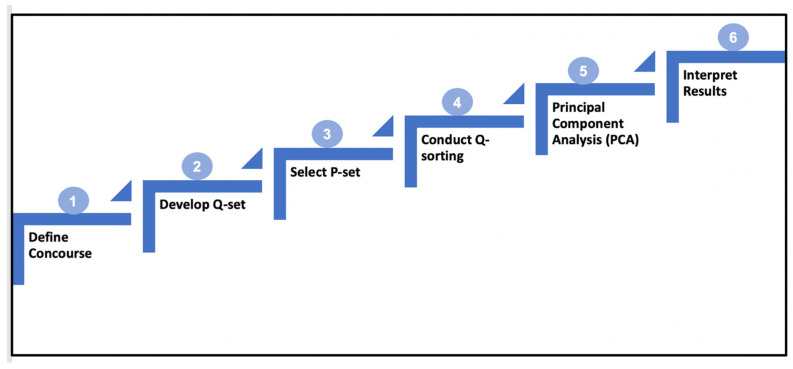
Q-methodology procedure.

**Table 1 animals-13-02930-t001:** Participants’ demographics.

	Experience (Years)	Organizational Role	Position	Relationship with PLF
P1	26–30	Veterinary services	Owner	PLF technology developer
		Technology development		
		Farming		
		Academic/research		
		Welfare certification/auditing		
P2	1–10	Welfare certification and regulation	Director animal care compliance	Potential off-farm PLF user
		Regulation		
P3	31 or more	Food retailing	Executive	Potential off-farm PLF user
		Welfare certification/auditing		
		Regulation		
		Education		
P4	1–10	Welfare certification and compliance	Consumer services manager	Potential off-farm PLF user
P5	11–15	Large-scale swine farming	Chief technology officer	PLF technology developer and farmer
		Technology development		
P6	16–20	Veterinary services	Director	PLF technology developer
		Technology development		
		Pharmaceutical		
P7	31 or more	Technology development	Director	PLF technology developer and farmer
P8	16–20	Swine veterinarians association	Director	Potential off-farm PLF user
P9	31 or more	Government regulation	Leader	Potential off-farm PLF user
P10	11–15	Academic/research institution	Senior researcher	PLF technology developer
		Technology development		
P11	31 or more	Academic/research institution	Professor	PLF technology developer
		Technology development		

**Table 2 animals-13-02930-t002:** By-person correlation matrix.

	P6	P4	P8	P3	P7	P9	P1	P10	P11	P2	P5
P6	1	−0.29	0.75	−0.38	0.73	0.28	0.58	0.64	0.24	0.50	0.60
P4		1	−0.21	0.59	−0.54	0.5	−0.46	−0.7	−0.3	−0.4	−0.50
P8			1	−0.25	0.46	0.43	0.58	0.64	0.34	0.50	0.52
P3				1	−0.62	0.17	−0.46	−0.18	−0.17	−0.17	−0.54
P7					1	0.17	0.55	0.62	0.14	0.24	0.68
P9						1	0.25	0.48	0.43	0.32	0.9
P1							1	0.49	0.25	0.50	0.56
P10								1	0.31	0.62	0.58
P11									1	0.28	0.4
P2										1	0.38
P5											1

**Table 3 animals-13-02930-t003:** More and least like what participants think of PLF.

Scores	Perspective 1	Perspective 2	Perspective 3
5	PLF should be a supplement to good management and not replace good management	PLF technology can disconnect caretakers from pigs and make them less caring about pigs	PLF usage is limited by poor internet connection on most swine farms
4	PLF will improve pig health	PLF technology is a poor proxy of a farmer looking over the herd, detecting problems, and addressing them	PLF usage can lead to data privacy and data ownership conflicts
4	PLF makes the pig caretaker’s job easier, safer, and better	PLF technology does not minimize the environmental impact of swine farming	PLF’s data often cannot be used within existing management practices
3	PLF will improve pig welfare	PLF technology may digitize swine production and make it look less natural and more artificial	PLF is cost-prohibitive to use across the entire livestock system
3	PLF will help in controlling disease outbreaks through traceability	PLF technology should be a supplement to good management and not replace good management	PLF requires a significant effort to change processes within an existing production system
−3	PLF is displacing labor and reducing job opportunities in the swine industry	PLF technology will make it possible for consumers to verify welfare certification claims	PLF is displacing labor and reducing job opportunities in the swine industry
−3	PLF often generates data that conflict with farmer expert opinions thus making PLF data less useful	PLF technology will address public concerns and increase consumer trust in pork production	PLF will address labor shortages in the swine industry
−4	PLF can be used without any form of training	PLF can be used without any form of training	PLF is a poor proxy of a farmer looking over the herd, detecting problems, and addressing them
−4	PLF can disconnect caretakers from pigs and make them less caring about pigs	PLF will improve pig health	PLF technology will address public concerns and increase consumer trust in pork production
−5	PLF’s data often cannot be used within existing management practices	PLF will improve pig welfare	PLF technology can be used without any form of training

**Table 4 animals-13-02930-t004:** Rotated solutions of participants’ Q-sorts.

Participants/Q-Sorts	Factor 1	Factor 2	Factor 3
P1	0.57 *	−0.51	0.2
P2	0.73 *	0.02	0.19
P3	−0.1	0.87 *	−0.03
P4	−0.01	0.86 *	−0.01
P5	0.62 *	−0.59	−0.17
P6	0.79 *	−0.37	0.08
P7	0.55 *	−0.68	−0.02
P8	0.76 *	−0.21	0.28
P9	0.46	0.22	0.63 *
P10	0.88 *	−0.08	0.17
P11	0.11	−0.13	0.91 *
% Eigenvalue	3.75	2.82	1.45
% Explained Variance	34	26	13

Factor loadings with an asterisk (*) are significantly associated with each factor. A sort was significantly associated with a factor at *p* < 0.01 statistical significance if the absolute value of the factor loading was greater than 0.47 [25].

## Data Availability

The data presented in this study are available on request from the corresponding author. The data are not publicly available to preserve participant identities.

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
