# Peer review of "US Swine Industry Stakeholder Perceptions of Precision Livestock Farming Technology: A Q-Methodology Study"

_animals, 2023, doi:10.3390/ani13182930_

Round 1

Reviewer 1 Report

General remark: very interesting study, filling a knowledge gap and challenging to further research.

Readability is good. I had to re-read some parts of the methodology description to get into the methodology, but that’s probably my lack of experience in the Q methodology.

Footnotes bothered me; they are not supporting fluent reading, so I would prefer to integrate them into the main text if possible.

Selecting the P-set. I understand that you want to test among those who gave input to the Q-set. However, I would be interested to also ask it to other stakeholders. This is not discussed.

The grouping of 55 statements into 30 sounds logic to me. However, which logic was used? Was the 30 an outcome or a goal, based on what? Some explanation here would be helpful.

Section 4 is mentioned Discussion, but is rather an elaboration of results per Perspective. Where is the Discussion? E.g. on the selected method, including having the same population who actually shaped the Q-set and tested it. It would make sense to widen the research to a larger population, including e.g. ngo’s and consumers.

Another discussion topic would be the question on whether the outcomes would be comparable to other countries, like EU or Asia; these have other farm sizes/dependency on workers, culture on use of data and of management information systems, and societal interest in animal welfare or labor conditions (to name a few). Even if it is not part of the research, a reference to potentially different implications elsewhere would add value (esp. given the limited P-set).

The discussion might contain a suggestion for further research towards different view points among (US) pig farmers.

In short: a very nice analysis, quite new, and relevant topic. For me it is necessary to replace the current content of section Discussion and add discussion topics.

Success in finalizing the paper.

Reviewer 2 Report

The article is well written and sufficiently argued the research questions. 

In the methodological part, the description part remains very weak, especially the justification of the use of the methodology and the selection of the sample. 

I recommend reading and citing the work of Dr Yari Vecchio (University of Bologna), who has critically addressed the use of the methodology for applications in the field of agricultural economics. 

The small sample size, which is not even half of the statements, should be better justified. 

I would like to read these improvements before making further changes.

Reviewer 3 Report

Review Swine industry stakeholder perception of precision livestock farming technology: A Q-methology study

Dear Authors, LPF is an intersting development in the swine industry. Therefor, this paper is of interest. By reading the paper i am missing important information to understand and interpret your results. I think this can be improved. See remarks. 

Title and further: Unfortunately it is not clear what the cultural origin is of the stakeholder. Based on the authors it can be American, but also European stakeholders. And as there are cultural and also swine industry differences this could have an effect on the results of the study. Please be more specific I the title, but more important in the Materials and Methods

L 41: consequently. this is not a result of increased consumption. It is a fact.

L42:  and further: Already here I would focus on the studied industry. As perception of welfare is culturally different.

L69-L70:  I don’t like the way other authors are referred to. Please make a sentence and then refer in the referrals. Now your sentence end really odd.  Hirtherto, very few…….exceptions are……[names].

L98: I also do not understand the note. Please explain fully in your materials and methods.

L125-132: I am puzzled by this part of the M&M. Actually, you are rephrasing your research question here. Which is not addressed is the abstract. Besides this remark I do not know whether using the same group for the Q set and the Q sorting is allowed in this Q-methodology.

L122: please no notes

L140: please no notes

Table 1: I am missing the consumer reported in line 116

From 164 you have lost me in the explanation. I do not understand what is done and how factors and perspectives are found.

As I don not understand the methodology, I cant review the results and the discussion.

 L 352 please write down the names of the authors you are referring to

Appendix 1

Some statements are missing: 2, 15, 20, 23, 26, 28. Why I do not understand.

Appendix 7 same problem, statements are missing.

I can’t review appendix 3-6 as I do not understand the methodology.

Round 2

Reviewer 2 Report

in the course of revisions I have not found substantial improvements to the article in order to see the work published. the justifications given by the authors for the lack of literature are weak and putting those articles as evidence for the number sampled is not sufficient. the article must be better long before it is published

Author Response

We appreciate your comments on our manuscript. However, we find your comment too generic, and it is not clear what you will have us do to improve the quality of the manuscript. It would help if you could clarify what you mean by the following statements:

  1. “I have not found substantial improvements to the article…” Are you referring to the entire article or some particular sections? We believe we have made very substantial changes to the article, reviewers 1 and 3 attested to these changes.
  2. “The justifications given by the authors for the lack of literature are weak and putting those articles as evidence for the number sampled is not sufficient” What justifications for lack of literature do you consider weak? If you refer to the lines where the justification for the lack of literature was made in the article this will help to address this comment.
  3. What do you mean by “putting those articles…” Are you referring to the peer-reviewed literature by Pilcher 2023; Hensel and Cifrino, 2023; Kenward et al., 2023; and Nezami et al., 2023 listed in our reviewer report round 1? If you consider citing a list of very recent peer-reviewed articles that support our methodology as not sufficient, then we would appreciate having your comments on what would suffice.
  4. We have no clue what you mean by “the article must be better long before it is published.”

Thank you for taking the time to review our manuscript and for your comments. We look forward to having your clarifications on these questions.  

Reviewer 3 Report

Hello,

Thank you for the new version and the alterations. 

Although I understand the massage of your paper, I still do not understand the method description of step 5. Also with the use of the reference (26) I still can not figure out what you are doing. line 188-224.

Author Response

Thank you for this constructive comment. We believe this comment will help a wider audience that is not familiar with Q methodology to understand our work. Please see the revised method description of step 5 in the new lines 188 – 231. We hope this clarifies your comments.
